# Graph Contrastive Learning with Stable and Scalable Spectral Encoding

**Deyu Bo**[1*], **Yuan Fang**[2], **Yang Liu**[1], **Chuan Shi**[1†]
[1]Beijing University of Posts and Telecommunications, China
[2]Singapore Management University, Singapore
{bodeyu, liuyangjanet, shichuan}@bupt.edu.cn, yfang@smu.edu.sg

## Abstract

Graph contrastive learning (GCL) aims to learn representations by capturing the agreements between different graph views. Traditional GCL methods generate views in the spatial domain, but it has been recently discovered that the spectral domain also plays a vital role in complementing spatial views. However, existing spectral-based graph views either ignore the eigenvectors that encode valuable positional information, or suffer from high complexity when trying to address the instability of spectral features. To tackle these challenges, we first design an informative, stable, and scalable spectral encoder, termed EigenMLP, to learn effective representations from the spectral features. Theoretically, EigenMLP is invariant to the rotation and reflection transformations on eigenvectors and robust against perturbations. Then, we propose a spatial-spectral contrastive framework ($\text{Sp}^2\text{GCL}$) to capture the consistency between the spatial information encoded by graph neural networks and the spectral information learned by EigenMLP, thus effectively fusing these two graph views. Experiments on the node- and graph-level datasets show that our method not only learns effective graph representations but also achieves a 2–10x speedup over other spectral-based methods.

## 1 Introduction

Graph neural networks (GNNs) have become the *de facto* framework to encode graph-structured data [14, 35, 39]. However, training high-quality GNNs usually requires a large number of domain-specific labels, which is not feasible in many real-world applications. Therefore, as a paradigm of self-supervised learning, graph contrastive learning (GCL) is proposed to learn node or graph representations without using labels [20, 19, 12, 38].

Typically, GCL methods first generate different views of a graph, and then contrast the positive views against the negative ones. By minimizing a contrastive loss, GCL methods can learn invariant information from different views for various downstream tasks. Therefore, how to generate ideal graph views is crucial to GCL. Most graph views are obtained by augmenting graphs in the spatial domain, such as dropping nodes and edges, heuristically [49, 44, 8] or adversarially [27, 40, 41]. Nevertheless, recent studies [18, 17] argue that spatial perturbations ignore the structural properties, and propose to perturb graph spectrum in the spectral domain.

Generally, the spatial and spectral views represent different information of the graph. The spatial domain captures the feature information and learns local graph representations by propagating the node features along local topology, *i.e.*, $k$-hop subgraphs. In contrast, the spectral domain covers the global structural information. The eigenvalues and eigenvectors encode the global shapes [13] and

---

[*]This work was done when the first author was a visiting student at Singapore Management University
[†]Corresponding author

node absolute positions [25]. Leveraging the agreements between the spatial and spectral views can significantly improve the expressive power and generalization ability of GNNs [37, 47]. However, several reasons hinder the study of spectral view: First, the spectral features are unstable. Previous work [24] shows that randomly flipping the signs or rotating the coordinates of eigenvectors also satisfies the eigenvalue decomposition, *a.k.a.*, the sign and basis ambiguity issues, implying that spectral features are not unique and hard to transfer [15, 37]. Besides, perturbing spectral features is time-consuming. Existing spectral-based methods [18, 17] need to decompose and reconstruct the adjacency matrix, in which the complexity is cubic and quadratic in the number of nodes, respectively.

Therefore, to realize the spatial-spectral contrast, it is natural to ask: *How to encode the spectral view of a graph effectively and efficiently?* To answer this question, we first propose a spectral encoder, named EigenMLP, which not only inherits the scalability advantage of multilayer perceptrons (MLP), but also adheres to two key design principles. First, EigenMLP resolves the sign ambiguity issue by taking both positive and negative eigenvectors as input. Second, to address the basis ambiguity issue, the weights of EigenMLP are generated by a learnable mapping of the eigenvalues, making them equivariant to the coordinates of eigenvectors. To integrate the representations learned from the spatial and spectral views, we propose a Spatial-Spectral GCL framework (Sp$^2$GCL) to maximize the agreements between views and learn effective representations for downstream tasks.

The contributions of our paper are as follows. (1) We propose EigenMLP, a novel spectral encoder to effectively and efficiently learn sign- and basis-invariant representations from spectral features. (2) We theoretically prove that EigenMLP is permutation-equivariant, rotation- and reflection-invariant, and can learn more stable representations against structural perturbations. (3) We propose Sp$^2$GCL, a spatial-spectral GCL framework that utilizes the cross-domain contrasts to effectively fuse the feature and structural information learned by spatial GNNs and EigenMLP. (4) Extensive experiments on both node-level and graph-level tasks demonstrate the effectiveness of the proposed framework Sp$^2$GCL, and verify the scalability and stability aspects of EigenMLP.

## 2 Related Work

**Graph Contrastive Learning.** Most GCL methods aim to learn invariant information by maximizing the mutual information between different graph views [1, 36]. There are many ways to generate graph views and we broadly categorize them into the spatial and spectral approaches. In the spatial domain, graph views are usually generated by augmenting the original graphs. For example, GRACE [48], GCA [49] and CCA-SSG [44] propose to augment graphs by randomly dropping edges and nodes, AD-GCL [27] leverages adversarial training to filter unimportant edges, and JOAO [41] combines different augmentation strategies automatically. While in the spectral domain, the views are generated by perturbing graph spectrum. For example, MVGRL [8] heuristically uses a graph diffusion as augmentation, which acts as a low-pass filter, SpCo [18] proposes to preserve low-frequency components and perturbs high-frequency ones, and SPAN [17] generates augmentations by maximizing the spectral change. Besides, SFA [46] analyzes the spectrum of node representations, which is out of the scope of this work. In general, spectral views of graphs may have better performance and interpretability than spatial views but also suffer from high complexity.

**Spectral Encoder.** Perturbing the graph spectrum is time-consuming. Another approach is to encode the eigenvalues and eigenvectors instead of the eigenspaces. However, eigenvectors suffer from sign and basis ambiguity issues, and using these features directly will affect the stability of the model. Therefore, some spectral encoders are proposed to learn invariant representations from the non-unique spectral features. For example, SAN [15] uses a Transformer-based [34] encoder, which is invariant to the order of the bases. BasisNet [16] uses IGN [21] to learn permutation-invariant representations from the eigenspaces. PEG [37] leverages the distance between eigenvectors to reweigh the graph structure and avoids sign and basis ambiguity. However, the complexity of SAN and BasisNet is quadratic, which is difficult to scale to large graphs.

## 3 Preliminaries

Assume that $\mathcal{G} = (\mathcal{V}, \mathcal{E})$ is a graph, where $\mathcal{V}$ is the node set with $|\mathcal{V}| = N$ and $\mathcal{E}$ is the edge set with $|\mathcal{E}| = E$. Let $\mathbf{A} \in \{0, 1\}^{N \times N}$ be the adjacency matrix, and $\mathbf{X} \in \mathbb{R}^{N \times d}$ be the node feature matrix on $\mathcal{G}$. The normalized graph Laplacian $\mathbf{L}$ of $\mathcal{G}$ is defined as $\mathbf{L} = \mathbf{I}_n - \mathbf{D}^{-\frac{1}{2}} \mathbf{A} \mathbf{D}^{-\frac{1}{2}}$, where $\mathbf{I}_n$ is the $N \times N$ identity matrix and $\mathbf{D}$ is the degree matrix with $D_{ii} = \sum_j A_{ij}$ for $i \in \mathcal{V}$ and $D_{ij} = 0$ for $i \neq j$. The eigenvalue decomposition (EVD) of graph Laplacian is defined as $\mathbf{L} = \mathbf{U} \mathbf{\Lambda} \mathbf{U}^\top$, where $\mathbf{\Lambda}$

is a diagonal matrix whose diagonal entries $0 \leq \lambda_1 \leq \cdots \leq \lambda_N \leq 2$ are the eigenvalues of $\mathbf{L}$, and $\mathbf{U} = [\mathbf{u}_1, \cdots, \mathbf{u}_N]$ are the corresponding eigenvectors.

It is worth noting that in some cases, randomly flipping the signs and rotating the coordinates of eigenvectors may also satisfy EVD [24], which we refer to as sign and basis ambiguity.

**Sign ambiguity.** Given a pair of eigenvalue and eigenvector $(\lambda_i, \mathbf{u}_i)$, it satisfies $\mathbf{L}\mathbf{u}_i = \lambda_i \mathbf{u}_i$, and $\lambda_i = \mathbf{u}_i^\top \mathbf{L}\mathbf{u}_i = \sum_{(v,v') \in \mathcal{E}} (u_{iv} - u_{iv'})^2$. Therefore, if $\mathbf{u}_i$ is an eigenvector of $\mathbf{L}$, then $-\mathbf{u}_i$ also satisfies EVD, *i.e.*, $\mathbf{u}_i^\top \mathbf{L}\mathbf{u}_i = (-\mathbf{u}_i)^\top \mathbf{L}(-\mathbf{u}_i)$.

**Basis ambiguity.** If there is high multiplicity in the eigenvalues, *i.e.*, $\lambda_{p+1} = \cdots = \lambda_{p+q}$ for some $q > 1$, then the corresponding eigenvectors $[\mathbf{u}_{p+1}, \cdots, \mathbf{u}_{p+q}]$ lie in an orthogonal group $\mathrm{O}(q)$ $= \{\mathbf{Q} \in \mathbb{R}^{q \times q} | \mathbf{Q}^\top \mathbf{Q} = \mathbf{Q}\mathbf{Q}^\top = \mathbf{I}_q\}$. Therefore, for any $\mathbf{Q} \in \mathrm{O}(q)$, replacing $[\mathbf{u}_{p+1}, \cdots, \mathbf{u}_{p+q}]$ with $[\mathbf{u}_{p+1}, \cdots, \mathbf{u}_{p+q}]\mathbf{Q}$ also satisfies EVD, *i.e.*, $\mathbf{L}\mathbf{u}_i = \lambda_j \mathbf{u}_i, p + 1 \leq i, j \leq p + q$.

The above two facts state that the eigenvectors of graph Laplacian are not unique. Therefore, the model should consider how to learn sign- and basis-invariant representations from spectral features for better stability and generalization [37, 16, 15].

# 4 Proposed Framework: Sp²GCL

In this section, we present the proposed spatial-spectral GCL framework called Sp²GCL. We first give a high-level overview on how to represent and contrast the spatial and spectral views of a graph. We then introduce the proposed invariant and equivariant spectral view encoder in detail. Finally, we briefly describe the preprocessing, training, and inference processes.

## 4.1 Overview of Sp²GCL

Views refer to different perspectives of the same data [30]. Unlike previous GCL methods that generate graph views in a single domain, we propose to model the spatial and spectral views separately and further utilize cross-domain contrasts to capture invariant information. Here we first describe how to represent the spatial and spectral views of graphs.

**Spatial View** represents the explicit connectivity of nodes, which can be denoted as $\mathbb{V}_a = (\mathbf{A}, \mathbf{X})$. Through propagating node features along graph substructures, the spatial view can naturally fuse the topology and content information and learn *local smooth representations* of a graph.

**Spectral View** indicates the implicit relationships between nodes, which is expressed as $\mathbb{V}_e = (\mathbf{\Lambda}, \mathbf{U})$. The eigenvalues and eigenvectors encode the geometric information and node positions of the graph topology, which can be seen as *global structural information* of a graph.

Since graphs are non-Euclidean data, it is difficult to directly contrast the spatial and spectral views. Therefore, we need to design suitable encoders to learn different view representations for GCL:

$$\mathbf{H}_a = f(\mathbf{A}, \mathbf{X}), \quad \mathbf{H}_e = g(\mathbf{\Lambda}, \mathbf{U}), \tag{1}$$

where $f$ and $g$ are the encoders of the spatial and spectral views, respectively, and $\mathbf{H}_a, \mathbf{H}_e \in \mathbb{R}^{N \times d}$ are the spatial and spectral representation matrices, respectively. These representations can then be used in contrastive learning to learn invariant information across both domains.

The fundamental idea of contrastive learning is to define the positive and negative pairs, from which the model can capture the self-supervised signals. In our framework, we define the spatial and spectral representations of the same node or graph as positive pairs, and those of different nodes or graphs as negative pairs. For graph-level contrasts, we additionally use a readout function to learn the graph-level representations. Then, two projection heads $\varphi_a, \varphi_e : \mathbb{R}^d \to \mathbb{R}^d$ are used to transform the representations into the contrastive space:

$$\mathbf{Z}_a = \varphi_a(\mathbf{H}_a), \quad \mathbf{Z}_e = \varphi_e(\mathbf{H}_e). \tag{2}$$

Subsequently, we employ InfoNCE [33], a classical contrastive objective function, to maximize the agreements between spatial and spectral representations:

$$\mathcal{L} = -\frac{1}{2} \sum_{i=1}^{N} \left( \log \frac{e^{\langle \mathbf{z}_a^i, \mathbf{z}_e^i \rangle}}{e^{\langle \mathbf{z}_a^i, \mathbf{z}_e^i \rangle} + \sum_{j \neq i} e^{\langle \mathbf{z}_a^j, \mathbf{z}_e^i \rangle}} + \log \frac{e^{\langle \mathbf{z}_e^i, \mathbf{z}_a^i \rangle}}{e^{\langle \mathbf{z}_e^i, \mathbf{z}_a^i \rangle} + \sum_{j \neq i} e^{\langle \mathbf{z}_e^j, \mathbf{z}_a^i \rangle}} \right), \tag{3}$$

Table 1: Comparison between different spectral encoders.

| Methods | Information | | Stability | | Scalability | |
| --- | --- | --- | --- | --- | --- | --- |
| | EigVal | EigVec | Sign | Basis | Inductive | Complexity |
| MLP [3, 4] | | ✓ | | | ✓ | $\mathcal{O}(Nkd)$ |
| SAN [15] | ✓ | ✓ | | ✓ | ✓ | $\mathcal{O}(Nk^2d + Nkd)$ |
| BasisNet [16] | | ✓ | ✓ | ✓ | | $\mathcal{O}(Nk^2d + Nkd)$ |
| PEG [37] | | ✓ | ✓ | ✓ | ✓ | $\mathcal{O}(Ekd)$ |
| EigenMLP | ✓ | ✓ | ✓ | ✓ | ✓ | $\mathcal{O}(NTd)$ |

*Note*: Here $k$ is the number of eigenvectors, $T$ is the period of polynomial, and $d$ is the hidden dimension.

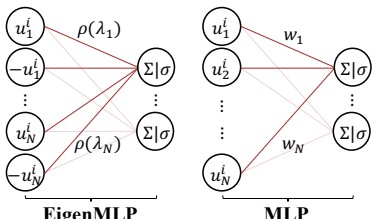

Figure 1: Schematic illustration of EigenMLP and MLP. [3]

where $\langle \cdot, \cdot \rangle$ represents the cosine similarity and $i$ is the index of nodes or graphs.

Finally, to materialize our contrastive framework Sp$^2$GCL, we must choose or design the encoders $f$ and $g$ for the two views. For the spatial encoder $f$, we directly employ a standard message-passing neural network (MPNN), *e.g.*, GraphSAGE [7] or GIN [39], which is widely adopted in previous GCL methods. For the spectral encoder $g$, which is the emphasis of this work, we propose EigenMLP in the next part (Section 4.2) based on several key properties. Note that, on the one hand, although MPNNs can learn useful representations, their expressive power is bounded by the 1-Weisfeiler-Lehman (WL) test [39]. On the other hand, the spectral view encodes the global information, which can help overcome the limitation of the 1-WL test [5]. Hence, from the perspective of expressive power, the spectral view is also complementary to the spatial view.

### 4.2 Proposed Spectral Encoder: EigenMLP

The spectral encoder is designed to learn stable representations from spectral features. A desirable spectral encoder should have three properties: (1) It can encode both the information of eigenvalues and eigenvectors, which represent different structural information [15]; (2) It is free of the sign and basis ambiguity issues in spectral features, toward learning stable representations [37]; (3) It is scalable to large graphs and has linear or sublinear time complexity. These three properties pose great challenges to the design of spectral encoders. Here we propose EigenMLP, which adopts an MLP-based architecture for scalability and addresses the sign and basis ambiguity problems for stability. Table 1 summarizes the differences between the proposed EigenMLP and existing spectral encoders. Specifically, EigenMLP satisfies all of the three properties above, as we elaborate below.

**Scalability.** In general, MLPs have demonstrated good scalability and have been widely used in learning graph representations [45, 11]. Since EigenMLP employs an MLP-based architecture, it inherits the low complexity and is linear to the number of nodes $N$.

**Stability.** MLPs are sensitive to input data, making the outputs vary with respect to the flipping of signs and rotation of coordinates of eigenvectors. This motivates us to increase the stability of MLP while maintaining high efficiency. We adopt two design principles that enable MLPs to learn sign- and basis-invariant representations. Figure 1 illustrates the difference between EigenMLP and MLP.

To address the sign ambiguity issue, we follow the design of SignNet [16], taking both positive and negative eigenvectors as inputs:

$$\tilde{\mathbf{U}} = [\psi(\phi(\mathbf{u}_i) + \phi(-\mathbf{u}_i))]_{i=1}^N, \tag{4}$$

where $\psi$ and $\phi$ are neural networks, $[\cdot]$ is the concatenation operator, and $\tilde{\mathbf{U}}$ is the sign-invariant eigenvectors. In practice, the sign-invariant neural networks may slow down model convergence. In this case, we can resort to some heuristics to determine the sign of the eigenvectors [47].

To solve the basis ambiguity issue, we first observe that each eigenvector has a corresponding eigenvalue, and when the coordinates of the eigenvectors are rotated, the positions of the eigenvalues are also displaced. Therefore, if we replace the weights of MLP with eigenvalues $\boldsymbol{\lambda} = [\lambda_1, \lambda_2, \cdots, \lambda_N]$, the model will be invariant to the rotation of eigenvectors, *i.e.*, $\mathbf{U}\mathbf{Q}(\boldsymbol{\lambda}\mathbf{Q})^\top = \mathbf{U}\mathbf{Q}\mathbf{Q}^\top\boldsymbol{\lambda}^\top = \mathbf{U}\boldsymbol{\lambda}^\top$. However, directly replacing the learnable weights with fixed eigenvalues is trivial and will greatly

---

[3]Some details, such as the neural networks $\phi$ and $\psi$, are omitted for brevity.

limit the expressive power of the model. Therefore, we first extend the scalar eigenvalues to their high-dimensional Fourier features [28]. The weights of EigenMLP are then decoded from the Fourier features using a learnable matrix:

$$\rho(\lambda) = [\sin(\lambda), \cos(\lambda), \sin(2\lambda), \cos(2\lambda), \cdots \sin(T\lambda), \cos(T\lambda)] \cdot \mathbf{W}_\rho, \tag{5}$$

where $T$ is the period and $\mathbf{W}_\rho \in \mathbb{R}^{2T \times d}$ is a learnable matrix. Here $\rho(\lambda)$ can be seen as a graph filter and the filtered eigenvalues are equivariant to the coordinates of eigenvectors [2], which can be used to learn powerful and basis-invariant spectral representations:

$$g(\mathbf{\Lambda}, \mathbf{U}) = \tilde{\mathbf{U}} \cdot \rho(\boldsymbol{\lambda}), \tag{6}$$

where $\rho(\boldsymbol{\lambda}) = [\rho(\lambda_1), \rho(\lambda_2), \cdots, \rho(\lambda_N)]^\top \in \mathbb{R}^{N \times d}$ represents all the filtered eigenvalues. The detailed matrix form of Equation (6) is provided in Appendix C. Note that the learnable matrix $\mathbf{W}_\rho$ is shared between different eigenvalues. Therefore, the size of $\mathbf{W}_\rho$ is independent of the number of eigenvalues but depends on the period $T$, which reduces the number of parameters in EigenMLP. More discussions can be found in Section 5.2.

**Information.** Notably, the invariant layer, *i.e.*, Equation (6), in EigenMLP incorporates both eigenvalues and eigenvectors, which can capture both geometric and positional information.

### 4.3 Preprocessing, Training, and Inference

It is worth noting that running EVD for full eigenvectors has the complexity $\mathcal{O}(N^3)$, which is unacceptable for large graphs. Therefore, in the preprocessing, we use the eigenvectors with smallest-$k$ eigenvalues as a substitute and reduce the complexity to $\mathcal{O}(N^2 k)$. Besides, we can pre-calculate the rotation-invariant spectral features $\mathbf{V} \cdot \rho(\boldsymbol{\lambda}_k)$. Therefore, in the training and inference, EigenMLP has the implementation of MLP. Note that the decomposition is only computed once in the training. Therefore, the overhead of the preprocessing should be amortized by each training epoch.

In training, traditional GCL methods need to compute the message-passing twice for different graph views, whose complexity is $\mathcal{O}(E)$. While in our framework, only the spatial view needs to calculate the message-passing, and the complexity of spectral view is $\mathcal{O}(N)$. Therefore, our framework is faster than others in the training processing. In inference, because we need to calculate the spatial and spectral representations separately, the inference speed is slightly lower than traditional methods. The overheads of training and inference are provided in Section 6.6.

## 5 Deeper Insights

In this section, we provide deeper insights into EigenMLP to understand its effectiveness. Specifically, we prove that EigenMLP is invariant, equivariant, and stable, and can generalize existing spectral augmentations. Besides, we also comment on the connections to previous work.

### 5.1 Theoretical Results

**Theorem 1.** *EigenMLP is equivariant to permutation, and invariant to rotation, and reflection.*

*Proof.* (Permutation) Assume that there are two matrices $\mathbf{L}^{(1)}, \mathbf{L}^{(2)} \in \mathbb{R}^{N \times N}$, and $\mathbf{L}^{(1)} = \mathbf{P}\mathbf{L}^{(2)}\mathbf{P}^\top$. We have $\mathbf{L}^{(1)} = \mathbf{P}\mathbf{L}^{(2)}\mathbf{P}^\top = (\mathbf{P}\mathbf{U}^{(2)})\mathbf{\Lambda}(\mathbf{P}\mathbf{U}^{(2)})^\top$ such that $\mathbf{U}^{(1)}\rho(\boldsymbol{\lambda}) = \mathbf{P}\mathbf{U}^{(2)}\rho(\boldsymbol{\lambda})$.

(Rotation) For any rotation matrix $\mathbf{Q} \in \mathbf{O}(N)$, the eigenvectors are rotated as $\mathbf{U}\mathbf{Q}$, and the corresponding Fourier matrix are permuted as $\mathbf{Q}^\top \rho(\boldsymbol{\lambda})$. Therefore, we have $\mathbf{U}\mathbf{Q} \cdot \mathbf{Q}^\top \rho(\boldsymbol{\lambda}) = \mathbf{U}\rho(\boldsymbol{\lambda})$.

(Reflection) Following Proposition 1 in [16], a continuous function is sign-invariant iff it satisfies $f(v) = h(v) + h(-v)$. Hence, EigenMLP is invariant to reflection on the signs of eigenvectors. $\square$

**Lemma 1.** *(Lemma 3.4 in [37]) For any positive semidefinite $\mathbf{L}$ without multiple eigenvalues, set $\mathbf{V}$ as the eigenvectors corresponding to the smallest $k$ eigenvalues and sorted as $0 < \lambda_1 < \cdots < \lambda_k$. For any sufficiently small $\epsilon > 0$, there exists a perturbation $\Delta\mathbf{L}$, $\|\Delta\mathbf{L}\|_F < \epsilon$ such that*

$$\min_{\mathbf{Q} \in \mathbf{O}(k)} \|(\mathbf{V} + \Delta\mathbf{V}) - \mathbf{V}\mathbf{Q}\|_F \geq 0.99 \max_{1 \leq i \leq k} |\lambda_{i+1} - \lambda_i|^{-1} \|\Delta\mathbf{L}\|_F + o(\epsilon). \tag{7}$$

**Theorem 2.** *For any sufficiently small $\epsilon > 0$, there exists a perturbation $\Delta\mathbf{L}$, $\|\Delta\mathbf{L}\|_F < \epsilon$ such that*

$$\forall_{\mathbf{Q}\in\mathbf{O}(k)} \|(\mathbf{V} + \Delta\mathbf{V})\rho(\boldsymbol{\lambda}_k) - \mathbf{V}\rho(\boldsymbol{\lambda}_k)\|_F \leq 0.99T \max_{1\leq i\leq k} |\lambda_{i+1} - \lambda_i|^{-1} \|\Delta\mathbf{L}\|_F + o(\epsilon). \quad (8)$$

Lemma 1 states that a small structural perturbation will produce changes unbounded from above in non-equivariant spectral features $\mathbf{V}$ if there is a small spectral gap. Theorem 2 shows that there is a finite upper bound on the changes of equivariant spectral features $\mathbf{V}\rho(\boldsymbol{\lambda}_k)$. Therefore, EigenMLP can learn more stable representations against structural perturbations. We present the proof of Theorem 2 in Appendix A. Experiments in Section 6.5 further give an empirical justification of Theorem 2.

**Proposition 1.** *EigenMLP generalizes existing spectral augmentations.*

*Proof.* Existing spectral augmentations, such as SpCo [18] and SPAN [17], generate graph views by perturbing the graph spectrum. We assume that there is a continuous univariate function $\delta(\cdot)$ between the original eigenvalues $\boldsymbol{\lambda}$ and the perturbed eigenvalues $\boldsymbol{\lambda}'$, such that $\lambda_i' = \delta(\lambda_i)$, and the perturbed eigenvalues are still non-negative. Then these spectral augmentations can be represented as:

$$\mathbf{A}' = \mathbf{U}\delta(\boldsymbol{\Lambda})\mathbf{U}^\top = (\mathbf{U}\delta(\boldsymbol{\Lambda})^{\frac{1}{2}})(\mathbf{U}\delta(\boldsymbol{\Lambda})^{\frac{1}{2}})^\top. \quad (9)$$

Note that Equation (5) is a Fourier series and can approximate any continuous functions. Therefore, the function $\delta(\cdot)$ is a special case of $\rho(\cdot)$, and EigenMLP can approximate the simplex geometry [32] of the augmentations, *i.e.*, $\mathbf{U}\delta(\boldsymbol{\Lambda})^{\frac{1}{2}}$, thus generalizing existing spectral augmentations. $\qquad\square$

### 5.2 Connections to Existing Work

**Hypernetworks** [6] are a class of neural networks used to generate parameters for another network. EigenMLP can be seen as a special case of Hypernetworks, which takes eigenvalues as input and generates equivariant parameters for spectral features. Because Hypernetworks can generate a set of non-shared weights from shared parameters, it can greatly reduce the number of trainable parameters. EigenMLP inherits this advantage, and its number of parameters is only related to the period of the Fourier series, rather than the number of eigenvectors. Therefore, EigenMLP is more efficient than other spectral encoders, including vanilla MLP.

**Spectral GNNs** aim to combine the eigenspaces with filtered eigenvalues, *i.e.*, $\sum_{i=1}^N \delta(\lambda_i)\mathbf{u}_i\mathbf{u}_i^\top$, and EigenMLP is used to combine the eigenvectors with filtered eigenvalues, *i.e.*, $\sum_{i=1}^N \rho(\lambda_i)\mathbf{u}_i$. Therefore, both methods choose to use eigenvalue mappings as weights to guarantee the equivariance. However, the calculation of eigenspaces has the complexity of $\mathcal{O}(N^2)$, which is not suitable for large graphs, whereas EigenMLP is more scalable.

## 6 Experiments

In this section, we conduct three types of experiments, including unsupervised node classification, unsupervised graph prediction, and transfer learning, to verify the effectiveness of Sp$^2$GCL. Besides, we also test the stability and time overhead of the proposed method.

### 6.1 Unsupervised Node Classification

**Datasets.** In the node classification task, we consider using graphs with different scales to evaluate both the effectiveness and scalability of GCL methods. Specifically, for the small graphs (< 50,000), we use Pubmed [14], Wiki-CS [22], and Facebook [23] datasets. For the large graphs (> 50,000), we use Flickr [43], arXiv [9], and PPI [7] datasets. Additional statistics are provided in Appendix B.

**Baselines and Setting.** We compare our model against a wide range of baselines, including semi-supervised GNNs, *e.g.*, GCN [14] and GAT [35], graph self-supervised learning methods, *e.g.*, DGI [36] and BGRL [29], GCL with spatial augmentations, *e.g.*, MVGRL [8], GRACE [48], and CCA-SSG [44], and GCL with spectral augmentations, *e.g.*, SpCo [18] and SPAN [17]. For the Facebook dataset, we randomly split the nodes into train/validation/test data with a ratio of 1:1:8. For other datasets, we use the public splits for a fair comparison. We use a two-layer GCN as the encoder for

Table 2: Node classification on transductive and inductive graphs. Mean accuracy (%) ± standard deviation. Bold indicates the best performance and "-" means out-of-memory or cannot be reproduced.

| Model | Data | Small Graphs (Full-Batch) | | | Large Graphs (Mini-Batch) | | |
|---|---|---|---|---|---|---|---|
| | | PubMed | Wiki-CS | Facebook | arXiv | Flickr | PPI |
| GCN | $\mathbf{A}, \mathbf{X}, \mathbf{Y}$ | 79.0 | 77.19±0.12 | 90.65±0.16 | 71.74±0.29 | 49.20±0.31 | 82.28±0.24 |
| GAT | $\mathbf{A}, \mathbf{X}, \mathbf{Y}$ | 79.0±0.3 | 77.65±0.11 | 90.47±0.15 | 71.82±0.23 | 54.48±0.21 | 98.85±0.05 |
| DGI | $\mathbf{A}, \mathbf{X}$ | 76.8±0.6 | 75.35±0.14 | 84.42±0.43 | 70.32±0.25 | 50.59±0.28 | 63.80±0.20 |
| BGRL | $\mathbf{A}, \mathbf{X}$ | 79.6±0.5 | 79.98±0.13 | 89.71±0.35 | 71.54±0.17 | 51.87±0.15 | 73.63±0.16 |
| MVGRL | $\mathbf{A}, \mathbf{X}$ | 80.1±0.7 | 77.52±0.08 | 87.29±0.28 | - | - | 71.45±0.14 |
| GRACE | $\mathbf{A}, \mathbf{X}$ | 80.6±0.4 | 80.14±0.48 | 89.32±0.40 | - | - | 69.71±0.17 |
| CCA-SSG | $\mathbf{A}, \mathbf{X}$ | 81.0±0.4 | 78.85±0.32 | 89.45±0.60 | 71.21±0.20 | 51.66±0.10 | 73.34±0.17 |
| SpCo | $\mathbf{A}, \mathbf{X}, \mathbf{\Lambda}$ | 81.5±0.4 | 79.16±0.27 | 89.98±0.45 | - | - | - |
| SPAN | $\mathbf{A}, \mathbf{X}, \mathbf{\Lambda}$ | 81.5±0.2 | **82.13±0.15** | - | - | - | - |
| Sp²GCL | $\mathbf{A}, \mathbf{X}, \mathbf{\Lambda}, \mathbf{U}$ | **82.3±0.3** | 79.42±0.19 | **90.43±0.13** | **71.83±0.19** | **52.05±0.33** | **74.28±0.22** |

Table 3: Results on the graph-level tasks. The best and runner-up results are highlighted with **bold** and underline, respectively. ↓ means lower the better, and ↑ means higher the better.

| Task | Regression (Metric: RMSE ↓) | | | Classification (Metric: ROC-AUC% ↑) | | | | |
|---|---|---|---|---|---|---|---|---|
| Dataset | molesol | mollipo | molfreesolv | molbace | molbbbp | molclintox | moltox21 | molsider |
| Supervised | 1.173±0.057 | 0.757±0.018 | 2.755±0.349 | 72.97±4.00 | 68.17±1.48 | 88.14±2.51 | 74.91±0.51 | 57.60±1.40 |
| InfoGraph | 1.344±0.178 | 1.005±0.023 | 10.005±4.819 | 74.74±3.64 | 66.33±2.79 | 64.50±5.32 | 69.74±0.57 | 60.54±0.90 |
| GraphCL | 1.272±0.089 | 0.910±0.016 | 7.679±2.748 | 74.32±2.70 | 68.22±1.89 | 74.92±4.42 | 72.40±1.01 | 61.76±1.11 |
| MVGRL | 1.433±0.145 | 0.962±0.036 | 9.024±1.982 | 74.20±2.31 | 67.24±1.39 | 73.84±4.25 | 70.48±0.83 | 61.94±0.94 |
| JOAO | 1.285±0.121 | 0.865±0.032 | 5.131±0.722 | 74.43±1.94 | 67.62±1.29 | 78.21±4.12 | 71.83±0.92 | 62.73±0.92 |
| AD-GCL | **1.217±0.087** | 0.842±0.028 | 5.150±0.624 | 76.37±2.03 | 68.24±1.47 | 80.77±3.92 | 71.42±0.73 | 63.19±0.95 |
| SPAN | 1.218±0.052 | **0.802±0.019** | 4.531±0.463 | 76.74±2.02 | **69.59±1.34** | 80.28±2.42 | 72.83±0.62 | **64.87±0.88** |
| Sp²GCL | 1.235±0.119 | 0.835±0.026 | **4.144±0.573** | **78.76±1.43** | 68.72±1.53 | 80.88±3.86 | 73.06±0.75 | 64.23±0.96 |

all datasets and set the hidden dimension $d = 512$ for all methods. For our model, the spatial encoder is the same as baselines, and we additionally use EigenMLP to learn the spectral representation. In the evaluation, we use a linear classifier to evaluate the performance of all methods, as suggested by [27]. We run all the models 10 times and report the mean accuracy and standard deviation. More details, *e.g.*, optimizers, and hyperparameters, are provided in Appendix B.

**Results.** From Table 2, we can find that Sp²GCL consistently outperforms state-of-the-art baselines on 5 out of 6 datasets, which validates the effectiveness of the proposed spatial-spectral contrastive framework. Meanwhile, spectral-based methods are proven to be more effective than spatial-based methods, suggesting that integrating spectral information into GCL can help models learn better representations. However, it is worth noting that neither SpCo nor SPAN works for large graphs, implying that perturbing graph spectrum cannot be scalable to large-scale datasets. Therefore, the application scenarios of these two graph augmentations are limited. On the contrary, our method can be used for large graphs and can be easily trained in a mini-batch manner, which is more scalable than other spectral-based methods. Additionally, We find that Sp²GCL does not perform well in the Wiki-CS dataset. The reason is that the node features dominate the classification results while the graph structure contributes less. Therefore, the spectral view cannot complement the spatial view.

## 6.2 Unsupervised Graph Prediction

**Setup.** We benchmark our model on the OGB graph property prediction task [9], which contains three regression datasets and five classification datasets. We choose a series of competitive GCL methods as baselines, including InfoGraph [26], GraphCL [42], MVGRL [8], JOAO [41], AD-GCL [27], and SPAN [17]. It is worth noting that SpCo [18] is not designed for graph-level contrastive learning, so we do not compare with it. We use a five-layer GIN [39] with a graph pooling layer as the encoder for all methods. Similarly, we additionally use EigenMLP to encode the spectral view for Sp²GCL. We use a linear downstream classifier, *e.g.*, logistic regression model, to evaluate the performance of different GCL methods, as suggested by [27].

Table 4: Graph transfer learning on the molecular classification tasks. (Metric: ROC-AUC (%) ↑)

| Dataset | ZINC-2M | | | | | | | |
|---|---|---|---|---|---|---|---|---|
| | BBBP | Tox21 | SIDER | ClinTox | BACE | HIV | MUV | ToxCast |
| No Pre-Train | 65.8±4.5 | 74.0±0.8 | 57.3±1.6 | 58.0±4.4 | 70.1±5.4 | 75.3±1.9 | 71.8±2.5 | 63.4±0.6 |
| InfoGraph | 68.8±0.8 | 75.3±0.5 | 58.4±0.8 | 69.9±3.0 | 75.9±1.6 | 76.0±0.7 | **75.3±2.5** | 62.7±0.4 |
| GraphCL | 69.7±0.7 | 73.9±0.7 | 60.5±0.9 | 76.0±2.7 | 75.4±1.4 | **78.5±1.2** | 69.8±2.7 | 62.4±0.6 |
| MVGRL | 69.0±0.5 | 74.5±0.6 | 62.2±0.6 | 77.8±2.2 | 77.2±1.0 | 77.1±0.6 | 73.3±1.4 | 62.6±0.5 |
| JOAO | **71.4±0.9** | 74.3±0.6 | 60.5±0.7 | **81.0±1.6** | 75.5±1.3 | 77.5±1.2 | 73.7±1.0 | 63.2±0.5 |
| AD-GCL | 70.0±1.1 | 76.5±0.8 | 63.3±0.8 | 79.8±3.5 | 78.5±0.8 | 78.3±1.0 | 72.3±1.6 | 63.1±0.7 |
| SPAN | 70.0±0.7 | 78.0±0.5 | **64.7±0.5** | 80.7±2.1 | 79.9±0.7 | 77.8±0.6 | 73.8±0.9 | 64.2±0.4 |
| Sp²GCL | 70.3±1.2 | **78.2±0.6** | 63.0±0.6 | 81.0±1.9 | 80.0±1.1 | 78.0±0.8 | 74.2±1.4 | **64.8±0.5** |

Table 5: Sp²GCL with different spectral encoders. Table 6: Justification of spatial-spectral contrast.

| | Pubmed | Facebook | Molbace |
|---|---|---|---|
| EigenMLP | 82.3 | 90.43 | 78.76 |
| MLP | 82.3 | 90.22 | 77.49 |
| SAN | 78.6 | 81.52 | 75.96 |
| BasisNet | 77.3 | 83.87 | 74.55 |

| | Pubmed | Facebook | Molbace |
|---|---|---|---|
| Sp²GCL | 82.3 | 90.43 | 78.76 |
| GRACE | 80.6 | 89.32 | 76.45 |
| +U | 81.2 | 89.85 | 77.32 |
| +U$\Lambda$ | 81.5 | 89.79 | 77.36 |

**Results.** Table 3 summarizes the graph prediction performance. Sp²GCL has 4 best and 3 runner-up performances and makes significant improvements on molfreesolv and molbace. Compared with GraphCL and JOAO, which employ multiple spatial augmentations, *e.g.*, edge and node dropping, and feature masking, Sp²GCL outperforms them by only using the spectral domain information. This demonstrates the effectiveness of spectral information. Compared with AD-GCL and SPAN, which use adversarial learning to find the near-optimal invariant information [31], Sp²GCL only uses the traditional InfoMax principle [1] and achieves competitive performance. This shows that the position information of spectral features is crucial to graph representation learning. Besides, in Table 9, we show that the efficiency and scalability of Sp²GCL significantly outperform adversarial learning.

## 6.3 Transfer Learning Scenario

**Setup.** The transfer learning experiments are used to evaluate the generalization ability of GCL methods. We follow the experimental setup described in [10], which involves pre-training the GCL methods on a large-scale dataset and subsequently fine-tuning the model on downstream datasets to evaluate their out-of-distribution performance We use the same graph encoder and baseline as in the unsupervised graph prediction task. Further details are provided in Appendix B.

**Results.** According to Table 4, Sp²GCL outperforms baselines in 4 out of 8 datasets and achieves an average rank of 1.7 across these datasets. Notably, the performance of different methods on different downstream datasets varies greatly, suggesting that the downstream tasks require distinct information for prediction. Nevertheless, spectral information proves to be crucial in the majority of tasks. Furthermore, Sp²GCL achieves comparable performance to SPAN, which employs a spectrum-based augmentation for learning graph representations. This observation indicates that Sp²GCL can learn similar spectrums as SPAN. However, our model falls short of InfoGraph, GraphCL, and JOAO on certain datasets due to the limited availability of feature information in the spectral view.

## 6.4 Ablation Studies

We conducted two ablation studies to validate the effectiveness of EigenMLP and the proposed framework, Sp²GCL. Initially, we replace EigenMLP with different spectral encoders and evaluate their performance on PubMed, Facebook, and molbace datasets. The results, presented in Tables 5, demonstrate that EigenMLP consistently outperforms other spectral encoders, thus confirming its effectiveness in learning spectral representations. Moreover, we aim to justify the superior ability of spatial-spectral contrast in fusing spatial and spectral representations. To accomplish this, we concatenate node features with eigenvectors and eigenvalues and subsequently feed them into GRACE. As shown in Table 6, we observe that incorporating spectral features improves the performance in

Table 7: Synthetic perturbations: Eigenvectors with random reflection $\mathbf{\Pi}$ and rotation $\mathbf{Q}$.

| PubMed | | Test | | |
|---|---|---|---|---|
| | | $\mathbf{U}$ | $\mathbf{U\Pi}$ | $\mathbf{UQ}$ |
| Train | $\mathbf{U}$ | 78.8 (+0.1) | 78.3 (-5.5) | 78.8 (-6.4) |
| | $\mathbf{U\Pi}$ | 78.5 (-7.8) | 78.9 (-0.4) | 79.0 (-5.2) |
| | $\mathbf{UQ}$ | 78.7 (-5.1) | 78.8 (-9.3) | 78.9 (+0.0) |

Table 8: Practical perturbations: Eigenvectors with different EVD tolerances.

| PubMed | | Test | | |
|---|---|---|---|---|
| | | $\mathbf{U}_3$ | $\mathbf{U}_4$ | $\mathbf{U}_5$ |
| Train | $\mathbf{U}_3$ | 78.6 (+0.3) | 73.9 (-1.5) | 73.7 (-7.8) |
| | $\mathbf{U}_4$ | 75.1 (-3.2) | 78.2 (+0.3) | 72.8 (-6.8) |
| | $\mathbf{U}_5$ | 73.7 (-6.9) | 72.0 (-5.0) | 79.7 (-0.5) |

Table 9: Time overheads (s) of different GCL methods.

| Method | Preprocessing | Training (100 Epochs) | Inference ($\times 10^{-3}$) |
|---|---|---|---|
| GRACE | 0.00 | 22.77 | 4.4 |
| MVGRL (GD) | 898.23 | 20.01 | 4.7 |
| SpCo (SI, $T$=10) | 127.16 | 22.81 | 4.5 |
| SPAN (EVD, $T$=10) | 658.62 | 142.92 | - |
| Sp$^2$GCL (EVD) | 66.44 | 17.79 | 6.2 |

Table 10: Overall time overheads (s) of different spectral encoders in 1000 forward passes.

| Method | Facebook ($k = 100$) | moltox21 (7831 Graphs) |
|---|---|---|
| MLP | 2.24 | 2.09 |
| SAN | 127.23 | 60.58 |
| BasisNet | 169.83 | 84.64 |
| EigenMLP | 5.34 | 3.14 |

GRACE. Nevertheless, the performance is still inferior to that of Sp$^2$GCL, thus providing further evidence of the effectiveness of spatial-spectral contrast.

### 6.5 Stability of EigenMLP and MLP

We conduct stability experiments to evaluate whether EigenMLP and MLP can learn stable representations against perturbations. We consider two types of perturbations: 1) Synthetic perturbation, which applies random reflection $\mathbf{\Pi}$ and rotations $\mathbf{Q}$ on the eigenvectors. 2) Practical perturbation, which decomposes the Laplacian matrix with different tolerances ($10^{-3}, 10^{-4}, 10^{-5}$). The corresponding eigenvectors are expressed as $\mathbf{U}_3$, $\mathbf{U}_4$, and $\mathbf{U}_5$. Note that synthetic perturbation only changes the signs and coordinates of eigenvectors while practical perturbation is more challenging because it perturbs the values. For clearer results, we only evaluate the performance of spectral representations.

For each type of perturbation, we construct three instances, *i.e.*, $(\mathbf{U}, \mathbf{U\Pi}, \mathbf{UQ})$ or $(\mathbf{U}_3, \mathbf{U}_4, \mathbf{U}_5)$. The models are trained on one instance and tested on the other two instances on PubMed. We report the results of EigenMLP and show the performance change after switching to MLP in the brackets. From Table 7, we can find that the synthetic perturbation can hardly change the representations learned by EigenMLP but has a great influence on MLP. Additionally, Table 8 shows that practical perturbations can affect both EigenMLP and MLP. Nevertheless, EigenMLP still consistently outperforms MLP across different tolerances, which verifies the stability of EigenMLP.

### 6.6 Time Overhead

To assess the efficiency, we compare the time overhead of Sp$^2$GCL with other GCL methods as well as compare EigenMLP with different spectral encoders. The time costs associated with preprocessing, training, and inference are shown in Table 9. Previous spectrum-based methods have significant time costs in the preprocessing stage, *e.g.*, graph diffusion (GD) or Sinkhorn's Iteration (SI), whereas Sp$^2$GCL exhibits minimal overhead. In the training stage, Sp$^2$GCL outperforms other methods due to the efficient spectral encoding, but it also introduces an additional burden during inference. Table 10 illustrates the overhead imposed by various spectral encoders within the Sp$^2$GCL framework. Both MLP and EigenMLP exhibit remarkable efficiency. Conversely, SAN and BasisNet entail excessive time costs due to their quadratic time complexity.

## 7 Conclusion

In this study, we introduce Sp$^2$GCL, a novel spatial-spectral GCL framework that learns the consistency between the spatial and spectral views of graphs. To effectively and efficiently learn the spectral view information, we propose EigenMLP, a scalable spectral encoder to learn stable spectral representations from the non-unique spectral features. Extensive experiments on various graph-related tasks demonstrate the effectiveness, efficiency, and stability of the proposed method.

**Limitation and Broader Impact**   Currently, we focus on encoding spectral features while ignoring node features. A promising future direction is to unify these two kinds of information to learn effective graph representations. Our work reveals the superiority of integrating spectral information into GCL and may inspire the community to pay more attention to the spectral view of graphs.

## Acknowledgments and Disclosure of Funding

This work is supported in part by the National Natural Science Foundation of China (No. U20B2045, 62192784, U22B2038, 62002029, 62172052). This work is also partially supported by the Ministry of Education, Singapore, under its Academic Research Fund Tier 2 (Proposal ID: T2EP20122-0041). Any opinions, findings and conclusions or recommendations expressed in this material are those of the author(s) and do not reflect the views of the Ministry of Education, Singapore.

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

# Appendices

## A Proof of Theorem 2

We use $i$ to indicate the index of $\lambda^{-1} = \max_{i \in N} |\lambda_{i+1} - \lambda_i|^{-1}$ and perturb graph Laplacian $\mathbf{L}$ by perturbing the eigenvectors. Specifically, we set

$$
\begin{aligned}
\mathbf{u}'_i =& \sqrt{1 - \epsilon^2}\mathbf{u}_i + \epsilon\mathbf{u}_{i+1}, \\
\mathbf{u}'_{i+1} =& -\epsilon\mathbf{u}_i + \sqrt{1 - \epsilon^2}\mathbf{u}_{i+1}.
\end{aligned}
\tag{1}
$$

Note that $||\mathbf{u}'_i|| = ||\mathbf{u}'_{i+1}|| = 1$ and $\mathbf{u}'^{\top}_i \mathbf{u}'_{i+1} = 0$. Therefore, replacing $\mathbf{u}_i$ and $\mathbf{u}_{i+1}$ with $\mathbf{u}'_i$ and $\mathbf{u}'_{i+1}$ still satisfies EVD. We denote $\mathbf{V}' = [\mathbf{u}_1, \cdots, \mathbf{u}'_i, \mathbf{u}'_{i+1}, \cdots, \mathbf{u}_k]$. Then the perturbation can be represented as $\Delta\mathbf{L} = \mathbf{V}'\mathbf{\Lambda}\mathbf{V}'^{\top} - \mathbf{V}\mathbf{\Lambda}\mathbf{V}^{\top} = \lambda_i(\mathbf{u}'_i\mathbf{u}'^{\top}_i - \mathbf{u}_i\mathbf{u}^{\top}_i) + \lambda_{i+1}(\mathbf{u}'_{i+1}\mathbf{u}'^{\top}_{i+1} - \mathbf{u}_{i+1}\mathbf{u}^{\top}_{i+1})$.

For sufficient small $\epsilon > 0$, we have:

$$
\begin{aligned}
&\min_{\mathbf{Q} \in \mathbf{O}(k)} ||(\mathbf{V} + \Delta\mathbf{V}) - \mathbf{V}\mathbf{Q}||_F \\
=& ||\mathbf{u}'_i - \mathbf{u}_i, \mathbf{u}'_{i+1} - \mathbf{u}_{i+1}||_F \\
=& ||(\sqrt{1 - \epsilon^2} - 1)\mathbf{u}_i + \epsilon\mathbf{u}_{i+1}||_F + ||(\sqrt{1 - \epsilon^2} - 1)\mathbf{u}_{i+1} - \epsilon\mathbf{u}_i||_F \\
=& 4(1 - \sqrt{1 - \epsilon^2}) \\
=& 2\epsilon^2 + o(\epsilon^2).
\end{aligned}
\tag{2}
$$

For Fourier features with period $\frac{T}{2}$, we have:

$$
\begin{aligned}
&\forall_{\mathbf{Q} \in \mathbf{O}(k)} ||(\mathbf{V} + \Delta\mathbf{V})\rho(\boldsymbol{\lambda}_k) - \mathbf{V}\rho(\boldsymbol{\lambda}_k)||_F \\
=& || \left[ \mathbf{u}'_i - \mathbf{u}_i, \mathbf{u}'_{i+1} - \mathbf{u}_{i+1} \right] [\rho(\lambda_i), \rho(\lambda_{i+1})]^{\top} ||_F \\
=& \sum_{t=1}^{T/2} || \sin(\lambda_i)(\mathbf{u}'_i - \mathbf{u}_i) + \sin(\lambda_{i+1})(\mathbf{u}'_{i+1} - \mathbf{u}_{i+1})||_F \\
&+ \sum_{t=1}^{T/2} || \cos(\lambda_i)(\mathbf{u}'_i - \mathbf{u}_i) + \cos(\lambda_{i+1})(\mathbf{u}'_{i+1} - \mathbf{u}_{i+1})||_F \\
\leq& T \left( ||\mathbf{u}'_i - \mathbf{u}_i||_F + ||\mathbf{u}'_{i+1} - \mathbf{u}_{i+1}||_F \right) \\
\leq& T(2\epsilon^2 + o(\epsilon^2)),
\end{aligned}
\tag{3}
$$

Next, we characterize $||\Delta\mathbf{L}||_F$:

$$
\begin{aligned}
&||\Delta\mathbf{L}||_F \\
=& \left\| \lambda_i \left( \mathbf{u}'_i\mathbf{u}'^{\top}_i - \mathbf{u}_i\mathbf{u}^{\top}_i \right) + \lambda_{i+1} \left( \mathbf{u}'_{i+1}\mathbf{u}'^{\top}_{i+1} - \mathbf{u}_{i+1}\mathbf{u}^{\top}_{i+1} \right) \right\|_F \\
=& \left\| (\lambda_{i+1} - \lambda_i) \left[ -\epsilon^2 \left( \mathbf{u}_i\mathbf{u}^{\top}_i - \mathbf{u}_{i+1}\mathbf{u}^{\top}_{i+1} \right) + \epsilon\sqrt{1 - \epsilon^2} \left( \mathbf{u}_i\mathbf{u}^{\top}_{i+1} + \mathbf{u}_{i+1}\mathbf{u}^{\top}_i \right) \right] \right\|^2_F \\
=& (\lambda_{k+1} - \lambda_k)^2 \left( \epsilon^2 \left\| \mathbf{u}_k\mathbf{u}^{\top}_{k+1} + \mathbf{u}_{k+1}\mathbf{u}^{\top}_k \right\|^2_F + o\left(\epsilon^2\right) \right) \\
=& 2 (\lambda_{k+1} - \lambda_k)^2 \left( \epsilon^2 + o\left(\epsilon^2\right) \right)
\end{aligned}
\tag{4}
$$

Combining Equations (2) and (4), we have the lower bound of the changes of non-equivariant spectral features under small perturbations:

$$
\min_{\mathbf{Q} \in \mathbf{O}(k)} ||(\mathbf{V} + \Delta\mathbf{V}) - \mathbf{V}\mathbf{Q}||_F \geq 0.99 \max_{1 \leq i \leq k} |\lambda_{i+1} - \lambda_i|^{-1} ||\Delta\mathbf{L}||_F + o(\epsilon),
\tag{5}
$$

which concludes the Lemma 1, *i.e.*, Lemma 3.4 in [37].

Combining Equations (3) and (4), we have the upper bound of the changes of equivariant spectral features under small perturbations:

$$
\forall_{\mathbf{Q} \in \mathbf{O}(k)} ||(\mathbf{V} + \Delta\mathbf{V})\rho(\boldsymbol{\lambda}_k) - \mathbf{V}\rho(\boldsymbol{\lambda}_k)||_F \leq 0.99T \max_{1 \leq i \leq k} |\lambda_{i+1} - \lambda_i|^{-1} ||\Delta\mathbf{L}||_F + o(\epsilon),
\tag{6}
$$

which concludes the Theorem 2.

# B    Detailed Experimental Setup

In this section, we report the details of our experiments. Specifically, we first introduce some general settings in all experiments. Then we introduce the detailed setup of each experiment one by one.

## B.1    General Settings

**Optimizer.**    For all experiments, we use the Adam optimizer.

**Environment.**    The environment in which we run experiments is:

- Linux version: 5.19.0-38-generic
- Operating system: Ubuntu 22.04.2
- CPU information: AMD EPYC 7313P 16-Core Processor
- GPU information: GeForce RTX 3090 (24 GB)

**Resources.**    The addresses and licenses of all datasets are as follows:

- PubMed: `https://github.com/tkipf/pygcn` (MIT License)
- Wiki-CS: `https://github.com/pmernyei/wiki-cs-dataset` (MIT License)
- Facebook: `https://github.com/benedekrozemberczki/MUSAE` (GPL-3.0 license)
- arXiv: `https://github.com/snap-stanford/ogb` (MIT license)
- Flickr: `https://github.com/GraphSAINT/GraphSAINT` (MIT license)
- PPI: `https://github.com/mims-harvard/ohmnet` (MIT license)
- OGB-graph: `https://github.com/snap-stanford/ogb` (MIT license)
- ZINC-2M: `https://github.com/snap-stanford/pretrain-gnns` (MIT license)

**Reproducibility.**    Our code is attached in the supplementary material.

## B.2    Unsupervised Node Classification

**Evaluation protocol.**    In the unsupervised node classification task, all methods are first trained with the corresponding self-supervised learning objectives. Then the learned representations are evaluated with a Logistic classifier with $l_2$ normalization. We evaluate the method every 10 epochs and the maximum epoch is set to 1000. For the mini-batch training, we set the batch size to 1024. The detailed statistics are shown in Table 1 and the hyperparameters are shown in Table 2.

Table 1: Statistics of unsupervised node classification datasets.

|  | **Graphs** | **Nodes** | **Edges** | **Features** | **Classes** |
|---|---|---|---|---|---|
| PubMed | 1 | 19,717 | 88,648 | 500 | 3 |
| Wiki-CS | 1 | 11,701 | 216,123 | 300 | 10 |
| Facebook | 1 | 22,470 | 342,004 | 128 | 4 |
| arXiv | 1 | 169,343 | 1,116,243 | 128 | 40 |
| Flickr | 1 | 89,250 | 899,756 | 500 | 7 |
| PPI | 24 | 56,928 | 1,226,368 | 50 | 121 |

## B.3    Unsupervised Graph Prediction

**Evaluation protocol.**    In the unsupervised graph prediction task, we use the stand encoder, provided by OGB [4], as the spatial encoder of Sp$^2$GCL, which is a 5-layer GIN with hidden dimension $d = 300$. We use add pooling to learn graph-level representations and set the batch size to 32. For the spectral

---

[4]https://github.com/snap-stanford/ogb/blob/master/ogb/graphproppred/mol_encoder.py

Table 2: Statistics of unsupervised node classification datasets.

|  | # Eigenvectors ($k$) | Period ($T$) | lr | wd | Dropout |
|---|---|---|---|---|---|
| PubMed | 30 | 20 | 1e-3 | 0 | 0 |
| Wiki-CS | 100 | 20 | 1e-3 | 0 | 0 |
| Facebook | 100 | 20 | 1e-3 | 0 | 0 |
| arXiv | 200 | 20 | 1e-3 | 0 | 0 |
| Flickr | 100 | 20 | 1e-3 | 0 | 0 |
| PPI | 50 | 20 | 1e-3 | 0 | 0 |

encoder, due to the relatively small sizes of the molecular graphs, we use all eigenvectors as the spectral features. We set the learning rate to 0.001 and the period to 10 for all datasets, and the number of training epochs is chosen among {20, 50, 80, 100, 150} using the validation set, as suggested by AD-GCL [27]. For the downstream evaluator, we use a Riger regressor for the regression tasks and a Logistic classifier for the binary classification tasks. The strength of $l_2$ normalization is grid searched among {0.001, 0.01, 0.1, 1, 10, 100, 1000} on the validation set for each dataset. The detailed statistics of the datasets are shown in Table 3.

Table 3: Statistics of unsupervised graph prediction datasets.

|  | Graphs | Avg. Nodes | Avg. Edges | Classes | Task | Metric |
|---|---|---|---|---|---|---|
| ogbg-molesol | 1,128 | 13.3 | 13.7 | 1 | Regression | RMSE |
| ogbg-mollipo | 4,200 | 27.0 | 29.5 | 1 | Regression | RMSE |
| ogbg-molfreesolv | 642 | 8.7 | 8.4 | 1 | Regression | RMSE |
| ogbg-molbace | 1,513 | 34.1 | 36.9 | 1 | Binary Class. | ROC-AUC |
| ogbg-molbbbp | 2,039 | 24.1 | 26.0 | 1 | Binary Class. | ROC-AUC |
| ogbg-molclintox | 1,477 | 26.2 | 27.9 | 2 | Binary Class. | ROC-AUC |
| ogbg-moltox21 | 7,831 | 18.6 | 19.3 | 12 | Binary Class. | ROC-AUC |
| ogbg-molsider | 1,427 | 33.6 | 35.4 | 27 | Binary Class. | ROC-AUC |

## B.4 Transfer Learning

**Evaluation protocol.** For the transfer learning task, we use the same GIN encoder as [10]. In the pre-training stage, the learning rate is set to 0.001 and the number of training epochs is chosen from {20, 50, 80, 100} based on the validation set. Similarly, we use all eigenvalues and eigenvectors as the spectral features, and the period is set to 10. In the fine-tuning stage, we remove the self-supervised learning objective, and an additional linear projection layer is used on the output of the encoder for classification. The hyperparameters are the same as in the pre-training stage. The detailed statistics of the datasets are shown in Table 4.

Table 4: Statistics of transfer learning datasets.

|  | Graphs | Utilization | Avg. Nodes | Avg. Edges |
|---|---|---|---|---|
| ZINC-2M | Pre-Training | 2,000,000 | 26.62 | 57.72 |
| BBBP | Finetuning | 2,039 | 24.06 | 51.90 |
| Tox21 | Finetuning | 7,831 | 18.57 | 38.58 |
| SIDER | Finetuning | 1,427 | 33.64 | 70.71 |
| ClinTox | Finetuning | 1,477 | 26.15 | 55.76 |
| BACE | Finetuning | 1,513 | 34.08 | 73.71 |
| HIV | Finetuning | 41,127 | 25.51 | 54.93 |
| MUV | Finetuning | 93,087 | 24.23 | 52.55 |
| ToxCast | Finetuning | 8,576 | 18.78 | 38.52 |

### B.5 Stability Experiment

We use the PyTorch-style pseudo code to explain how we generate the synthetic perturbations (Figure 2) and practical perturbations (Figure 3).

```
e, u = torch.linalg.eigh(L) # EVD

random_sign = 2*torch.randint(0, 2, (N,))- 1
sign_flip = torch.diag(random_sign).float()
coor_flip = torch.randperm(N)

u_sign = torch.mm(u, sign_flip)
u_basis = u.clone()[:, coor_flip]
```

```
e3, u3 = scipy.sparse.linalg.eigsh(
    L, k=100, which='SM', tol=1e-3)

e4, u4 = scipy.sparse.linalg.eigsh(
    L, k=100, which='SM', tol=1e-4)

e5, u5 = scipy.sparse.linalg.eigsh(
    L, k=100, which='SM', tol=1e-5)
```

Figure 2: Synthetic perturbations        Figure 3: Practical perturbations

## C    Matrix Form of EigenMLP

We give a detailed matrix form of Equation 6, from which we can see that the Fourier features of eigenvalues give different weights to the eigenvectors, thus making the model invariant to the rotation of coordinates and preserving good fitting ability.

$$
\underbrace{\begin{bmatrix} u_1^1 & u_2^1 & \cdots & u_k^1 \\ u_1^2 & u_2^2 & \cdots & u_k^2 \\ \vdots & \vdots & \ddots & \vdots \\ u_1^N & u_2^N & \cdots & u_k^N \end{bmatrix}}_{\text{Eigenvectors, } N \times k} \times \underbrace{\begin{bmatrix} \cos(\lambda_1) & \sin(\lambda_1) & \cdots & \cos(T\lambda_1) & \sin(T\lambda_1) \\ \cos(\lambda_2) & \sin(\lambda_2) & \cdots & \cos(T\lambda_2) & \sin(T\lambda_2) \\ \vdots & \vdots & \vdots & \ddots & \vdots & \vdots \\ \cos(\lambda_k) & \sin(\lambda_k) & \cdots & \cos(T\lambda_k) & \sin(T\lambda_k) \end{bmatrix}}_{\text{Fourier features of eigenvalues, } k \times 2T}
$$

$$
\times \underbrace{\begin{bmatrix} \alpha_1^1 & \alpha_2^1 & \cdots & \alpha_d^1 \\ \alpha_1^2 & \alpha_2^2 & \cdots & \alpha_d^2 \\ \vdots & \vdots & \ddots & \vdots \\ \alpha_1^{2T} & \alpha_2^{2T} & \cdots & \alpha_d^{2T} \end{bmatrix}}_{\text{Parameters of learnable matrix, } 2t \times d} = \underbrace{\begin{bmatrix} h_1^1 & h_2^1 & \cdots & h_d^1 \\ h_1^2 & h_2^2 & \cdots & h_d^2 \\ \vdots & \vdots & \ddots & \vdots \\ h_1^N & h_2^N & \cdots & h_d^N \end{bmatrix}}_{\text{Representations, } N \times d} \tag{7}
$$

## D    Pseudo Algorithm

In order to better demonstrate our algorithm, here we provide the pseudo algorithms of EigenMLP (Figure 4) and Sp$^2$GCL (Figure 5).

Figure 4: Pseudo Algorithm of EigenMLP

```python
class EigenMLP(nn.Module)

    def __init__(self, k, d, T):
        # k: number of Eigenvectors
        # d: hidden dimension
        # T: period
        self.phi = nn.Sequential(nn.Linear(1, d), nn.ReLU(), nn.Linear(d, d))
        self.psi = nn.Sequential(nn.Linear(d, d), nn.ReLU(), nn.Linear(d, 1))
        self.mlp = nn.Sequential(nn.Linear(2*T, d), nn.ReLU(), nn.Linear(d, d))

    def forward(e, u):
        u = u.unsqueeze(-1)
        u = self.psi(self.phi(u) + self.phi(-u)).squeeze(-1)          # [N, k]

        T_term = torch.arange(0, T).float()
        T_e = e.unsqueeze(1) * T
        F_e = torch.cat([torch.sin(T_e), torch.cos(T_e)], dim=-1)   # [k, 2T]

        return self.mlp(torch.mm(u, F_e))
```

Figure 5: Pseudo Algorithm of Sp$^2$GCL

```python
def Sp2GCL(g, x, e, u):
    # g: graph structures
    # x: node features
    # e: eigenvalues
    # u: eigenvectors
    x_a = GNN(g, x)
    x_e = EigenMLP(e, u)

    # For graph-level tasks
    # x_a = add_pool(g, x_a)
    # x_e = add_pool(g, x_e)

    h_a = spa_projection_head(x_a)
    h_e = spe_projection_head(x_e)

    h_a = F.normalize(h_a, dim=-1, p=2)
    h_e = F.normalize(h_e, dim=-1, p=2)

    logits = torch.mm(h_a, h_e.t())
    labels = torch.arange(h_a.size(0), dtype=torch.long)

    return 0.5 * F.cross_entropy(logits, labels) +
            0.5 * F.cross_entropy(logits.t(), labels)
```

