# OpenReview forum: "Graph Contrastive Learning with Stable and Scalable Spectral Encoding"
_NeurIPS.cc/2023/Conference — NeurIPS 2023 poster_

### Official Review · Reviewer_ieZM · 2023-06-28

**Soundness:** 2 fair
**Presentation:** 2 fair
**Contribution:** 2 fair
**Rating:** 6
**Confidence:** 4

**Summary:**

The paper proposes a method for Graph Contrastive Learning (GCL) by contrasting the spatial and spectral views ($Sp^2GCL$). The spatial view is obtained using a message passing GNN. For the spectral view the authors propose an equivariant model called EigenMLP. EigenMLP precomputes the k smallest eigenvalues and corresponding eigenvectors for a graph. The eigenvectors are made sign invariant using positive and negative eigenvectors as in SignNet. Permutation/Basis equivariance is obtained by learning MLP weights from fourier features of the eigenvalues. The method uses the same node/graph representations as positive views and from different graph/nodes as negative views. The InfoNCE contrastive function is used as the objective. After the self supervised training, a linear classifier is used on the downstream task of node/graph classification/prediction.

**Strengths:**

1) Paper proposes an important direction of combining spatial and spectral views
2) The method obtains competitive results compared with baselines


**Weaknesses:**

Regarding the views for a node/graph, the spectral views are obtained from the eigendecomposition that are made equivariant to sign and basis and the spatial views are obtained from an MPNN. This would give a fixed view for every node/graph and the concern is for a node/graph how to obtain multiple positive views? In the absence of multiple views the learning may be limited to fixed representations and may not scale well for larger models.

**Minor Typos:**
1. The eigenvalues and eigenvectors **encoder** the global shapes [13] and node absolute positions $\rightarrow$ The eigenvalues and eigenvectors **encode** the global shapes [13] and node absolute positions
2. It can **encoder** both the information of eigenvalues and eigenvectors $\rightarrow$ It can **encode** both the information of eigenvalues and eigenvectors
3. In practice, the sign-invariant neural networks may slow down model **converge** $\rightarrow$ In practice, the sign-invariant neural networks may slow down model **convergence**
4. Therefore, EigenMLP can learn more stable representations **again** structural perturbations $\rightarrow$ Therefore, EigenMLP can learn more stable representations **against** structural perturbations


**Questions:**

1. Regarding the views for a node/graph, the spectral views are obtained from the eigendecomposition that are made equivariant to sign and basis and the spatial views are obtained from an MPNN. This would give a fixed view for every node/graph and the concern is for a node/graph how to obtain multiple positive views? In the absence of multiple views the learning may be limited to fixed representations and may not scale well for larger models.
2. As pointed out by the authors, the spatial method learns local features and the spectral method learns global properties. Is it always the right approach to contrast them in the proposed manner? In which cases would it work and how do practitioners make a decision?


**Limitations:**

Some of the limitations have been addressed in the paper. Please refer to the Weakness and Questions section to address further limitations.

---

> ### Author Rebuttal · Authors · 2023-08-08
>
> Thanks for your positive comments!
>
> > **Q1: How to obtain multiple positive views? In the absence of multiple views the learning may be limited to fixed representations and may not scale well for larger models.**
>
> A1: Constructing multiple positive views is important for contrastive learning. To achieve this goal, the most widespread approach is to augment the input data.
>
> Several approaches exist for augmenting spatial features [1][2], including randomly dropping edges, nodes, and features. On the other hand, for spectral features, we can create multiple positive views by selecting different numbers of eigenvectors. For example, in descending order of eigenvalues, we can select the first $k$, $2k$, ..., eigenvectors. Each of these views captures distinct frequencies of node positions, thereby enabling the construction of multi-scale representations.
>
> [1] Graph Contrastive Learning with Augmentations. NeurIPS 2020.
>
> [2] Graph Contrastive Learning Automated. ICML 2021.
>
> > **Q2: Minor Typos**
>
> A2: Thank you for carefully checking our paper. We will polish our paper based on your suggestions in the revision.
>
> > **Q3: Is it always the right approach to contrast them in the proposed manner? In which cases would it work and how do practitioners make a decision?**
>
> A3: The definition of positive and negative views, as well as the selection of contrastive objective function, continue to be open challenges in the field of contrastive learning. Consequently, spatial-spectral contrast may not always be the right approach, depending on the property of the data.
>
> Intuitively, the spatial methods encode the local feature information through message-passing and the spectral methods learn the positional information. Therefore, spatial-spectral contrast works well **if the positional information can complement the feature information**. However, it may not yield favorable results if the feature information dominates the classification accuracy. For example, in some heterophilic datasets, where two connected nodes tend to have different labels, the performance of MLP is better than GNNs [3]. In such cases, the positional information may not complement the feature information.
>
> [3] Graph Neural Networks for Graphs with Heterophily: A Survey

---

> > ### Comment · Reviewer_ieZM · 2023-08-13
> > **Response to rebuttal**
> >
> > Thanks to the authors for the clarifications and it helps my understanding of the proposed method. I will think through the discussed points in detail and consider my review in light of the responses.
> >
> > Many Thanks

---

> > > ### Author Response · Authors · 2023-08-14
> > > **Thanks for your response**
> > >
> > > Hi, we are pleased to receive your prompt reply. If you have any questions, we are willing to discuss and clarify.
> > >
> > > Best regards.

---

### Official Review · Reviewer_QJUG · 2023-07-04

**Soundness:** 2 fair
**Presentation:** 2 fair
**Contribution:** 2 fair
**Rating:** 4
**Confidence:** 3

**Summary:**

The authors present a novel approach called Sp2GCL that combines spatial and spectral views of graphs using EigenMLP, an informative and stable spectral encoder. The proposed method shows promising results in learning effective graph representations and outperforms other spectral-based methods in terms of both performance and efficiency.

**Strengths:**

This work proposes contrasting two views in the spectral domain and spatial domain and introduces a novel encoder called EnigenMLP to encode spectral domain information, which has not been done by former work.

**Weaknesses:**

- The contribution of the article is considered limited as the traditional Graph Neural Network (like GCN), which can be understood as spectral filtering in the spectral domain, is similar to the EnigenMLP proposed in this work.

- Simply contrasting these two representations might not lead to significant improvements, as indicated in the results where there is no clear enhancement and suspicion of cherry-picking.

- The article suggests that the proposed method may not offer significant improvement over Graph Convolutional Network methods, and the results indicate a lack of noticeable enhancement. This raises concerns about the effectiveness of the proposed approach.
The article mentions that the contribution of the work is relatively limited, indicating that the proposed method may not introduce significant advancements beyond existing approaches.


**Questions:**

- How are the scalar eigenvalues extended to high-dimensional Fourier features in EigenMLP?
- What are the overheads of training and inference in EigenMLP?
- How does EigenMLP handle the sign and basis ambiguity issues in spectral features?

---

> ### Author Rebuttal · Authors · 2023-08-08
>
> Thanks for your helpful suggestions!
>
> > **Q1: The contribution of the article is considered limited as the traditional Graph Neural Network (like GCN), which can be understood as spectral filtering in the spectral domain, is similar to the EnigenMLP proposed in this work.**
>
> A1: EigenMLP is fundamentally different from the spectral filtering applied in Graph Neural Networks (GNNs) for several reasons:
>
> 1. EigenMLP serves as a method for encoding the **positional information** of nodes. In contrast, spectral filtering of GNNs utilizes graph spectrum to filter the noise present in **node features**.
>
> 2. The expressive power of traditional GNNs is limited by the  Weisfeiler-Lehman test [1]. In contrast, incorporating the positional information can surpass this limitation [2]. Therefore, the expressive power of EigenMLP is better than message-passing GNNs.
>
> [1] How powerful are graph neural networks? ICLR 2019.
>
> [2] Graph neural networks with learnable structural and positional representations. ICLR 2022.
>
> > **Q2: Simply contrasting these two representations might not lead to significant improvements, as indicated in the results where there is no clear enhancement and suspicion of cherry-picking.**
>
> A2: We make an additional experiment to validate the effectiveness of the spatial-spectral contrast. Specifically, we directly concatenate the spatial features $A^{2}X$ and the spectral features $U\rho(\Lambda)$ as non-contrastive representations, i.e., $[A^{2}X, U\rho(\Lambda)]$. Then we use a  linear classifier, which is the same as Sp2GCL, to evaluate the performance of non-contrastive representations. The experiment is conducted in the Pubmed and Flickr datasets. Results are shown below, from which we can see that the spatial-spectral contrast contributes a lot to learning graph representations.
>
> | | Contrastive | Non-contrastive |
> :-: | :-: | :-:
> | Pubmed | 82.3±0.3 | 80.1±0.2 |
> | Flickr | 52.05±0.33 | 50.27±0.47 |
>
> > **Q3: The article suggests that the proposed method may not offer significant improvement over Graph Convolutional Network methods, and the results indicate a lack of noticeable enhancement. This raises concerns about the effectiveness of the proposed approach. The article mentions that the contribution of the work is relatively limited, indicating that the proposed method may not introduce significant advancements beyond existing approaches.**
>
> A3: We need to clarify that graph convolutional networks (GCNs) are **semi-supervised methods**, implying that they need label supervision to update the model parameters. In contrast, our model is an **unsupervised method** that does not require the usage of labels. Simply comparing the performance of GCNs and our method is unfair. Moreover, we can see that in some datasets, our model outperforms GCNs a lot. For example, in the Pubmed dataset, our method (accuracy=82.3) has an improvement of 4% over GCNs (accuracy=79.0), which demonstrates the effectiveness of the proposed method.
>
> Additionally, we do not mention that the contribution of our work is relatively limited. We consistently highlight that our model is an effective and efficient method.
> - In terms of effectiveness, our model achieves comparable performance over various graph-related tasks.
> - In terms of efficiency, Sp2GCL is 10 times faster than the state-of-the-art spectral-based GCL method, and EigenMLP is 30 times faster than existing sign- and basis-invariant spectral encoders.
>
> > **Q4: How are the scalar eigenvalues extended to high-dimensional Fourier features in EigenMLP?**
>
> A4: *We describe the Fourier features of eigenvalues in Equation 5.*
>
> Specifically, we stack the sin and cos values of the scalar eigenvalues with different periods to construct its high-dimensional Fourier features, i.e., $[\sin(\lambda), \cos(\lambda), \sin(2\lambda), \cos(2\lambda), \cdots \sin(T\lambda), \cos(T\lambda)]$.
>
> > **Q5: What are the overheads of training and inference in EigenMLP?**
>
> A5: *The time overheads of EigenMLP and Sp2GCL are shown in Tables 9 and 10.*
>
> Here we briefly report the results for a quick reference. In summary, Sp2GCL is 10 times faster than the state-of-the-art spectral-based GCL method, e.g., SPAN, and EigenMLP is 30 times faster than existing sign- and basis-invariant spectral encoders.
>
> | Table 9 | Pre-processing | Training (100 epochs) | Inference ($\times 10^{-3}$) |
> :-: | :-: | :-: | :-:
> | SpCo | 127.16 | 22.81 | 4.5 |
> | SPAN | 658.62 | 142.92 | - |
> | Sp2GCL | 66.44 | 17.79 | 6.2 |
>
> | Table 10 | Facebook (k = 100) | moltox21 (7831 Graphs) |
> :-: | :-: | :-:
> | MLP | 2.24 | 2.09 |
> | SAN | 127.23 | 60.58 |
> | BasisNet | 169.83 | 84.64 |
> | EigenMLP | 5.34 | 3.14 |
>
> > **Q6: How does EigenMLP handle the sign and basis ambiguity issues in spectral features?**
>
> A6: *We describe how EigenMLP solves the sign and basis ambiguities in Section 4.2. We also theoretically prove this in Theorem 1, Section 5.1.*
>
> - For sign-ambiguity, EigenMLP simultaneously takes the positive and negative eigenvectors as input, thus learning sign-invariant representations.
> - For basis-ambiguity, EigenMLP leverages the learned eigenvalues to weight eigenvectors, thus eliminating the influence of coordinate rotation.

---

### Official Review · Reviewer_xYrm · 2023-07-07

**Soundness:** 4 excellent
**Presentation:** 3 good
**Contribution:** 3 good
**Rating:** 5
**Confidence:** 5

**Summary:**

In this paper, the authors propose eigenMLP,an informative, stable, and scalable spectral encoder, which is invariant to the rotation and reflection transformations on eigenvectors and robust against perturbations. Based on eigenMLP,  spatial-spectral contrastive framework is proposed to capture the consistency between the spatial information and spectral information.

**Strengths:**

1. The eigenMLP is very motivated. The motivation and the theoretical analysis are convincing.

2. Based on eigenMLP, the spectral augmentation is bounded by pertubation delta L.



**Weaknesses:**

There are some major issues:
1. I think the eigenMLP is the main contribution but I don't know why the authors try to highlight that the spatial-spectral contrastive framework is a contribution. In my opinion, it is not the first work using spectral-based GNN and spatial-based GNN as two different views for GCL.

2. Because of the sparsity, eigenvalues usually obtain by randomized SVD. The computational cost is nk^2 rather than n^2k usually. I guess you do not use SVD designed for dense matrices. If you do you can check randomized SVD that may help you.

3. The novelty of eigenMLP: the eigenMLP looks like SignNet + position embedding.

**Questions:**

1. What is the T in Eq.8?

2. EigenMLP is able to outperform other spectral-based GNNs with the usual semi-supervised setting?

3. Why the MLP is so close to EIgenMLP in Table 5?

---

> ### Author Rebuttal · Authors · 2023-08-08
>
> Thanks for your positive comments!
>
> > **Q1: I think the eigenMLP is the main contribution but I don't know why the authors try to highlight that the spatial-spectral contrastive framework is a contribution.**
>
> A1: We agree that the main contribution of our paper is the design of EigenMLP, which is an effective and efficient spectral encoder. On the other hand, most GCL methods focus on a single domain, i.e., spatial or spectral, and fewer methods work on both domains. Therefore, we list the spatial-spectral contrastive framework as our contribution. In the revision, we will discuss the difference between our model and existing two-view GCL methods.
>
> > **Q2: If you do you can check randomized SVD that may help you.**
>
> A2: We express gratitude for your valuable feedback. Subsequent testing proves that random SVD can further reduce the complexity of the preprocessing. Coupled with the fact that our model is efficient during the training phase, the whole process is more scalable and efficient.
>
> Specifically, we compare the sparse eigenvalue decomposition (EVD) algorithm (numpy.linalg.eigh) and the randomized SVD algorithm (sklearn.utils.extmath.randomized_svd). The outcomes demonstrate a notable acceleration in computation. For instance, in the context of the Pubmed dataset, the EVD procedure takes approximately **70 seconds**, while the SVD process is accomplished in nearly **2 seconds**.
>
> > **Q3: The novelty of eigenMLP: the eigenMLP looks like SignNet + position embedding.**
>
> A3: Generally, both SignNet and EigenMLP can be seen as positional encoding methods. The novelty of EigenMLP comes from two perspectives:
>
> 1. SignNet can only address the sign-ambiguity problem, while its advanced version, BasisNet, comes with a heavy computational burden. In contrast, EigenMLP presents an efficient and effective approach that simultaneously tackles the sign- and basis-ambiguity issues. As demonstrated in Table 10, EigenMLP is 30 times faster than BasisNet.
>
> 2. EigenMLP provides an efficient way to perturb the graph spectrum for better contrastive learning. Existing spectral augmentation methods, e.g., SpCo [1] and SPAN [2], need to decompose and reconstruct the graph structure, which is inefficient. EigenMLP can directly learn new eigenvalues from the Fourier features of raw eigenvalues. However, as we can see from Table 1, both SignNet and BasisNet cannot utilize the eigenvalues.
>
> > **Q4: What is the T in Eq.8?**
>
> A4: The symbol $T$ is the period of the Fourier features, which is a hyper-parameter in our model.
>
> > **EigenMLP is able to outperform other spectral-based GNNs with the usual semi-supervised setting?**
>
> A5: We conduct the semi-supervised experiment in the Pubmed dataset because it has the standard semi-supervised data split, i.e., 20 nodes per class for training and 1000 randomly sampled nodes for test [1]. We choose three competitive spectral GNNs as baselines: GPR-GNN [2], ChebyNet [3], and BernNet [4]. Notably, EigenMLP does not use feature information, while spectral GNNs use graph spectrum to filter the node features. Therefore, to perform a fair comparison, we use two different settings:
>
> 1. **Pubmed w/ features**: In this setting, we concate the representations learned by EigenMLP and original node features, and feed them into an MLP for classification. Spectral GNNs remain unchanged.
>
> 2. **Pubmed w/o features**: In this setting, we replace the node feature matrix with an identity matrix, and feed it into spectral GNNs for classification. As for EigenMLP, we only use the eigenvectors and eigenvalues.
>
> The results are shown as follows, from which we can see that EigenMLP has a great improvement over spectral GNNs in learning positional information. Remarkably, even with the inclusion of node features, EigenMLP still maintains its superiority over the baselines.
>
> | | Pubmed w/ features |Pubmed w/o features |
> :-: | :-: | :-:
> | EigenMLP | **80.15±0.43** | **75.62±0.16** |
> | GPR-GNN | 79.92±0.38 | 71.42±0.12 |
> | ChebyNet | 78.53±0.26 | 52.44±0.38 |
> | BernNet | 79.86±0.24 |61.76±0.45 |
>
> [1] Semi-Supervised Classification with Graph Convolutional Networks. ICLR 2017.
>
> [2] Adaptive Universal Generalized PageRank Graph Neural Network. ICLR 2021.
>
> [3] Convolutional Neural Networks on Graphs with Fast Localized Spectral Filtering. NeruIPS 2016.
>
> [4] BernNet: Learning Arbitrary Graph Spectral Filters via Bernstein Approximation. NeurIPS 2021.
>
> > **Q6: Why the MLP is so close to EIgenMLP in Table 5?**
>
> A6: In Table 5, Pubmed and Facebook are transductive datasets, while Molbace is non-transductive. In transductive datasets, the training, validation, and test nodes share the same adjacency matrix. Consequently, the signs and coordinates of eigenvectors remain consistent during the training and inference stages. Therefore, the model performance will not be influenced by sign- and basis-ambiguity. As a result, the performance of MLP is close to that of EigenMLP in Pubmed and Facebook. On the contrary, EigenMLP outperforms MLP a lot in the Molbace dataset.
>
> The same phenomenon can also be observed in the stability experiment (Section 6.5). When the training and test data employ the same perturbation, the performance of MLP is comparable to that of EigenMLP, as evident from the results in the diagonal entries of Tables 7 and 8. However, in cases where different perturbations are applied, EigenMLP consistently outperforms MLP, as indicated by the results in the off-diagonal entries.

---

### Official Review · Reviewer_9zn6 · 2023-07-07

**Soundness:** 3 good
**Presentation:** 2 fair
**Contribution:** 2 fair
**Rating:** 5
**Confidence:** 4

**Summary:**

This paper proposes a graph contrastive learning model with spatial and spectral augmentations, with a novel spectral encoder EigenMLP that could address the stability issue from eigen-decomposition. To exploit the strength of spatial and spectral domains, SP2GCL deploys two augmentation views for the contrastive framework; in the spectral view, it introduces tricks to alleviate the ambiguity of signs and basis, in order to stabilize the training process. The performances of node-level and graph-level experiments, as well as the transfer learning tasks, show the advantages of the proposed framework.

**Strengths:**

1. Contrastive learning for graph data is a prosperous and promising field, especially for the label-sparse settings and for the exploration of properties of non-Euclidean data.
2. The spectral part in the proposed framework attempts to address the stability and overhead issues, which are the main obstacle in the spectral methods for graphs.
3. Bridging among the spatial and spectral views is an interesting attempt, which could also make full use of their complementary properties.

**Weaknesses:**

1. As the authors emphasize the stability of their method, more theoretical and empirical analysis are expected to validate it.
2. I feel confused about the introduction of Fourier features in Line 169; the reasons and influences of them could be more detailed.
3. The performances are not always competitive in the experiment section; it may help to explain based on the properties of data sets.

**Questions:**

In the 3rd paragraph of Introduction, it's stated that spatial methods capture local features and so as spectral ones to global features, does it still true for the augmentations? Is it possible that some spectral augmentations merely perturb some kind of edges (such as low-degree or deviant ones)?

---

> ### Author Rebuttal · Authors · 2023-08-09
>
> Thanks for your valuable suggestions!
>
> > **Q1: As the authors emphasize the stability of their method, more theoretical and empirical analysis are expected to validate it.**
>
> A1: Here are the theoretical and empirical analyses:
> 1. **Theoretical analysis**: In Section 5.1, we theoretically analyze the stability of our method against structural perturbations. Specifically, Theorem 2 states that the inverse of the spectral gap bounds the change of EigenMLP, while Lemma 1 shows that the change of MLP is unbounded. Therefore, EigenMLP is more stable than MLP.
>
> 2. **Empirical analysis**: In Section 6.5, we conduct an experiment to validate the theoretical analysis, where two types of perturbations are used to evaluate the stability of EigenMLP and MLP. Detailed results are shown in Tables 7 and 8.
>
> **To further verify the adversarial robustness**, we conduct a new experiment that injects adversarial edges into the graph structures and tests the performance of Sp2GCL and other GCL methods on the Pubmed dataset. We consider three competitive baselines, GRACE [1], SPAN [2], and SP-AGCL [3].
>
> Specifically, we consider two graph adversarial attacks, i.e., Nettack [4] and Metattack [5]. To perform a fair comparison, we employ the data generated by Pro-GNN [6]. The training, validation, and testing nodes are randomly divided in a ratio of 1:1:8. The results are shown below, where Sp2GCL consistently learns stable representations against various adversarial attacks, while other GCL methods are more vulnerable.
>
> | | Clean | Nettack ($n=5$) | Metattack ($p=0.2$) |
> :-: | :-: | :-: | :-:
> | GRACE | **85.9±0.1** | 74.5±1.3 ($\downarrow$13.3%) | 71.4±0.2 ($\downarrow$16.9%) |
> | SPAN | 81.5±0.3 | 76.4±2.1 ($\downarrow$6.3%) | 72.7±0.6 ($\downarrow$10.8%) |
> | SP-AGCL | 85.5±0.3 | 78.1±1.6 ($\downarrow$8.7%) | 75.1±0.5 ($\downarrow$12.2%) |
> | Sp2GCL | 83.3±0.5 | **79.1±1.7 ($\downarrow$5.0%)** | **75.4±0.4 ($\downarrow$9.5%)** |
>
> [1] Deep Graph Contrastive Representation Learning.
>
> [2] Spectral Augmentation for Self-Supervised Learning on Graphs. ICLR 2023.
>
> [3] Similarity Preserving Adversarial Graph Contrastive Learning. KDD 2023.
>
> [4] Adversarial Attacks on Neural Networks for Graph Data. KDD 2018.
>
> [5] Adversarial Attacks on Graph Neural Networks via Meta Learning. ICLR 2019.
>
> [6] Graph Structure Learning for Robust Graph Neural Networks. KDD 2020.
>
> > **Q2: I feel confused about the introduction of Fourier features in Line 169; the reasons and influences of them could be more detailed.**
>
> A2: Before introducing the role of Fourier features, we first explain the role of eigenvalues. Eigenvalues are instrumental in addressing the basis-ambiguity issue as they are equivariant to the rotation of eigenvectors, i.e., $\mathbf{U}\mathbf{Q} \cdot (\Lambda\mathbf{Q})^{\top} = \mathbf{U} {\Lambda}^{\top}$, where $\mathbf{Q}$ is the random rotation matrix.
>
> Notably, the raw eigenvalues tend to assign greater significance to higher frequencies. However, in practice, low-frequency information also wields considerable importance [7]. Employing raw eigenvalues falls short of attaining optimal results. To overcome this limitation, it becomes necessary to learn a set of new eigenvalues to reweight the eigenvectors.
>
> In this paper, we choose to learn new eigenvalues from the Fourier features of the raw eigenvalues, i.e., $λ_{new}=[\sin(λ), \cos(λ), ..., \sin(Tλ), \cos(Tλ)]\mathbf{W}$. This design has been widely used in other fields. For example, Transformer uses positional encoding to preserve the order information of input tokens [8].
>
> The advantages of Fourier features are two-fold:
> - Fourier features provide a multi-scale representation of scalar eigenvalues and let neural networks learn high-frequency information [9].
> - Fourier features are bounded in [-1, 1], which are more stable than other methods, such as polynomials.
>
> [7] Revisiting Graph Neural Networks: All We Have is Low-Pass Filters.
>
> [8] Attention Is All You Need. NeurIPS 2017.
>
> [9] Fourier Features Let Networks Learn High Frequency Functions in Low Dimensional Domains. NeurIPS 2020.
>
> > **Q3:The performances are not always competitive in the experiment section; it may help to explain based on the properties of data sets.**
>
> A3: In some chemical datasets, the performance of EigenMLP drops slightly. This may be due to the fact that the properties of the molecule are determined by some important substructures [10] rather than the global structure. However, eigenvectors encode the global position information and are not good at modeling substructures. A possible solution is to use data augmentation to mask some nodes and obtain the eigenvectors of subgraphs.
>
> [10]  Convolutional networks on graphs for learning molecular fingerprints. NeurIPS 2015.
>
> > **Q4: Does it still true for the augmentations? Is it possible that some spectral augmentations merely perturb some kind of edges (such as low-degree or deviant ones)?**
>
> A4: We think this statement still holds for graph augmentations. According to the definition of eigenvalue decomposition, the graph structure is composed of different eigenspaces, i.e., $\mathbf{A}=\mathbf{U} \Lambda \mathbf{U}^{\top} = \sum \lambda_{i} u_{i} {u_{i}}^{\top} $.
> Because the eigenvector $u_{i}$ is a dense vector, the induced eigenspace $u_{i} {u_{i}}^{\top}$ is a dense matrix. Therefore, a small perturbation in the eigenvalues will result in a global change in the graph structure, i.e., $\Delta \mathbf{A} = \Delta \lambda_{i} u_{i} {u_{i}}^{\top}$.

---

> > ### Comment · Reviewer_9zn6 · 2023-08-18
> >
> > Thanks for the clarification from authors. I have carefully read the rebuttal, which addressed most of my concerns. I will raise my score.

---

> > > ### Author Response · Authors · 2023-08-19
> > > **Thanks for your appreciation**
> > >
> > > We express our sincere gratitude for your response. Thanks for your appreciation of our paper.

---

### Author Rebuttal · Authors · 2023-08-10

We extend our gratitude to all the reviewers for their valuable feedback and insightful suggestions. We have diligently addressed the majority of the questions and suggestions raised during the official review process, and have provided comprehensive responses to individual reviewers in the corresponding rebuttals.

We are delighted to be recognized for our efforts in this research. We would like to express our appreciation to Reviewer 9zn6 for acknowledging the novelty and significance of our model in the context of spectral-based methods. Our thanks also go to Reviewer xYrm for endorsing our motivation and theoretical contributions. We thank Reviewer QJUG for the helpful suggestions. Lastly, we would like to convey our gratitude to Reviewer ieZM for their strong acknowledgment of the novelty and effectiveness of our work.

---

### Decision · Program_Chairs · 2023-09-21

**Decision:**

Accept (poster)

**Comment:**

The paper introduces a new spectral encoder for graphs, and apply it to contrastrive learning to achieve demonstrably competitive accuracy at a fraction of the computational cost if compared with other spectral methods. The paper received overall positive reviews, highlighting the relevance of the problem, the novelty of the proposed spectral encoder, and the idea of combining spatial and spectral views in one method. Some concerns on the adversarial robustness and missing comparisons with semi-supervised methods were adequately addressed in the rebuttal with additional quantitative results, together with a deeper theoretical discussion. The paper is a solid contribution in the graph learning area, and is a step forward in bridging spatial and spectral graph learning methods.